# Facile Fabrication of Transparent and Opaque Albumin Methacryloyl Gels with Highly Improved Mechanical Properties and Controlled Pore Structures

**DOI:** 10.3390/gels8060367

**Published:** 2022-06-10

**Authors:** Mengdie Xu, Nabila Mehwish, Bae Hoon Lee

**Affiliations:** 1Wenzhou Institute, University of Chinese Academy of Sciences, Wenzhou 325011, China; xumd@wiucas.ac.cn (M.X.); nabila@wiucas.ac.cn (N.M.); 2Oujiang Laboratory, (Zhejiang Lab for Rengerative Medicine, Vision and Brain Health), Wenzhou 325000, China

**Keywords:** bovine serum albumin methacryloyl, cryogels, hydrogels, porous biomaterials, mechanical properties

## Abstract

For porous protein scaffolds to be employed in tissue-engineered structures, the development of cost-effective, macroporous, and mechanically improved protein-based hydrogels, without compromising the original properties of native protein, is crucial. Here, we introduced a facile method of albumin methacryloyl transparent hydrogels and opaque cryogels with adjustable porosity and improved mechanical characteristics via controlling polymerization temperatures (room temperature and −80 °C). The structural, morphological, mechanical, and physical characteristics of both porous albumin methacryloyl biomaterials were investigated using FTIR, CD, SEM, XRD, compression tests, TGA, and swelling behavior. The biodegradation and biocompatibility of the various gels were also carefully examined. Albumin methacryloyl opaque cryogels outperformed their counterpart transparent hydrogels in terms of mechanical characteristics and interconnecting macropores. Both materials demonstrated high mineralization potential as well as good cell compatibility. The solvation and phase separation owing to ice crystal formation during polymerization are attributed to the transparency of hydrogels and opacity of cryogels, respectively, suggesting that two fully protein-based hydrogels could be used as visible detectors/sensors in medical devices or bone regeneration scaffolds in the future.

## 1. Introduction

A globular heart-shaped “Bovine serum albumin (BSA)” is a simple and cost-effective protein with appealing properties, including biodegradation and biocompatibility, which has been extensively reported for drug delivery and extended for therapeutic and medicinal uses, including ocular distribution [1,2,3,4,5]. It is one component of the tissue-culture media, and a scaffold based on albumin should work well for cell and tissue engineering because it can provide structural support [6,7,8]. It has also been suggested as a biomaterial scaffold for in-vivo implants [9] and in-vitro culture of fibroblasts, osteoblasts, and other cells [10,11,12,13]. With good cell viability, photocurable albumin hydrogels demonstrated tunable physiological characteristics over swelling, mechanical properties, and degradation [14,15,16]. Moreover, albumin hydrogels have exhibited valuable properties for regenerative medicine and tissue engineering because of their inertness, reduced immunogenicity, low cost, biodegradability, and the ability for creating patient-specific albumin [17]. As a result of its appealing qualities, BSA has extensive opportunities for its functionalization and preparation of diverse biomaterials aimed at tissue repair applications. Considering the abundance and availability of albumin, just a few alternative synthetic approaches have been established so far, [18,19,20,21] and it has not been broadly explored. Therefore, better methods for manufacturing and functionalizing albumin hydrogels are required.

Various tough transparent hydrogels have been synthesized as smart windows, as antibacterial coatings, and as optical detectors for tissue engineering and industrial applications [22,23,24,25,26,27,28,29,30,31,32]. Similarly tough opaque hydrogels have been utilized more often as bone regeneration or bone tissue engineering scaffolds [33]. For example, polyampholyte-based tough hydrogels have been added to poly(*N*-isopropylacrylamide) (PNIPAAm) for a changeable shift to clear and opaque [27]. For manufacturing smart windows, transparency is an essential requisite for ensuring indoor lighting and contribution by solar energy; consequently, pillar[6]arene and ferrocene were employed to generate a warm and cool tone reversible thermochromic substance [23]. Different crosslinking densities have been used to make cellulose gels, which have been proven to generate hydrogels with varying transparency and mechanical strength [34]. The effects of a variety of simple carbohydrates on the optical transparency, mechanical characteristics, and pore size of hyaluronan (HA) gels were demonstrated [35]. It is well known that diffusion of nutrients, as well as cell proliferation and ECM excretion, are all controlled by the porosity of hydrogels, with pore size regulating cell growth [36]. Furthermore, including a porous structure in the hydrogel can significantly improve swelling behavior [21]. As a result, it is realistic to expect that hydrogels with a porous structure will have more applications in tissue engineering [37,38,39]. However, there have been scarcely any reports on the design and synthesis of bio-compatible, bio-degradable, and bio-mineralizable (3B) cost-effective transparent and opaque protein-based gels with highly improved mechanical properties and controlled pore structure.

Herein, porous transparent and opaque protein-based hydrogels are generated from one cost-effective protein building block i.e., modified BSA (bovine serum albumin methacryloyl—BSAMA) under different gelation temperatures and keeping all other factors constant. To manufacture cost-effective porous transparent and opaque gels, free radical polymerization at room, and sub-zero, temperature was utilized (Figure 1). Swelling, degradation, mechanical characteristics, and cell viability of BSAMA hydrogels and cryogels were compared in a systematic way. Using BSAMA and free radical polymerization, the presented research proposes a facile and cost-effective system of making transparent and opaque BSAMA gels with improved mechanical properties and controllable pore structures to be potentially employed as useful substrates for tissue engineering in the future.

## 2. Results and Discussion

BSAMA was synthesized following the earlier reports [11,16] (Figure 1a) and was characterized by proton NMR (^1^HNMR) to confirm the conjugation of methacryloyl to BSA. BSA and BSAMA NMR spectra exhibit relatively broad peaks, and some peaks are overlapped; a qualitative analysis of NMR results is presented here. As presented in Figure 1b, compared to the peaks of BSA, new specific peaks at about 5.4 and 5.7 ppm were found on the BSAMA spectrum (peak i, ii), equivalent to the acrylic protons (2H) of the methacrylamide, and a broad new peak (peak iv) at about 1.9 ppm corresponding to the methyl protons (3H) of the methacrylamide. MAA seemed to react completely with the free amino groups available in BSA, and the methylene proton (2H) peak of lysine (peak iii) was shifted from about 3.0 ppm to 3.2 ppm after the methacryloylation. Since the specific peaks (especially at around 6.1 ppm) of methacrylate groups were absent, the resultant BSAMA was found to mainly contain methacrylamide groups [16,40].

Circular dichroism (CD) was employed [41] for the secondary structure of the BSA and BSAMA. The CD spectrum of natural BSA has two negative bands at 208 and 222 nm, which is characteristic of α-helix proteins (Figure 1c) [42]. While BSAMA exhibited a similar CD pattern as that of BSA, both negative bands decreased in intensity slightly possibly due to the insertion of methyl and acryloyl groups; the α-helical content of BSA molecules decreased slightly during the methacryloylation [16]. BSA has a higher fractional helicity (53.81%) in comparison to BSAMA (45.88%) at RT (Figure 1d).

In order to formulate the tough scaffolds, higher polymer concentrations (10%, 15%, and 20% (*w*/*v*)) were used in accordance with the literature [11,16]. To check the kinetics of free radical polymerization, the tilt-tube test was utilized at RT to identify the sol-gel phase transition with reference to time. Since a lower concentration usually takes more time to gel due to a smaller number of polymer chains available in the solution, we have shown the time-dependent gelation of 10% BSAMA at RT (Appendix A). It can be seen that polymerization starts within 30 min and that a stable gel was obtained at around 12 h; so as to make this study comparable, the same polymerization duration was also applied to cryogelation. Cryogels with different pore sizes can be made by changing several factors like the freezing temperature, the polymer concentration, and the crosslinker content [43]. Therefore, to optimize the cryogelation temperature, BSAMA cryogelation was tested at two different temperatures (−20 and −80 °C). As shown in Appendix A, cryogels obtained at −20 °C appeared not homogenous, and the pore morphology did not show a clear trend among samples with different concentrations possibly because the cryogelation at −20 °C might be not sufficient to support fast phase separation and radical polymerization simultaneously. On the other hand, cryogels obtained at −80 °C seemed more homogeneous and more stable and exhibited a more defined macroporous pore morphology compared to those at −20 °C. This result is consistent with the previous report [44]. Therefore, the freezing temperature of −80 °C was chosen for cryogelation of BSAMA.

Accordingly, BSAMA hydrogels and cryogels have been fabricated via free radical polymerization at RT and −80 °C, respectively. As shown in Figure 1 and Appendix A, both the transparent and opaque gels were prepared by redox-induced polymerization in the presence of APS/TEMED [45,46]. Interestingly, the hydrogels formed at RT were transparent while the cryogels (hydrogels prepared at −80 °C) were opaque. This phenomenon was consistent for all the concentrations tested (Appendix A). 10% (*w*/*v*) hydrogels appeared soft and fragile (Appendix A) compared to higher concentrations tested. It has been reported that dense polymer networks with high cross-linking formed with the increase in the polymer concentration [47] and eventually more stable gels were formed.

Fourier transform infrared spectroscopy (FTIR) can help understand material structure, hence, as presented in Figure 2, BSA and BSAMA were analyzed and compared with FTIR spectra of BSAMA hydrogels and cryogels. In the case of natural BSA (Figure 2a), characteristic amide I and amide II peaks appeared at 1648 cm^−1^ and 1535 cm^−1^, respectively, along with amide A band around 3270 cm^−1^ confirming the secondary structure of the protein [16]. BSAMA along with the transparent and opaque gels showed similar characteristic peaks of the BSA illustrating the maintenance of the protein structure during the MAA functionalization and free radical polymerization. Amide I band was deconvoluted and curve fitted (Figure 2b–e and Appendix A) for the analysis of individual secondary structures of the proteins [16,48]. BSA, BSAMA, and opaque BSAMA gels showed similar percent α-helical content (~50%), while the hydrogels contained less α-helix structure (~40%), and more unidentified structures (irregular coils). There was no change in the inclusive form of amide I band which shows that all samples remain similar to α- helix secondary structure of BSA.

As shown in Figure 3, the morphology and microstructure of the as-prepared scaffolds (hydrogels/cryogels) were analyzed by SEM (cross-section of the lyophilized gels, in Figure 3a), and CLSM (surface of the hydrated dye-stained gels in Figure 3b). Even though the interior structure/morphology of the dried gels after freeze-drying would be different from the natural state of the swollen gels, SEM analyses can be supportive of elucidating the 3D internal structure of the gels [49]. The cross-sectional internal morphology of the freeze-dried hydrogels showed a porous character with smaller pores with a diameter in the range of approximately 10–60 µm, compared to cryogels with a larger pore of 70–200 µm) at all the concentrations tested (Figure 3a). Under the influence of polymerization conditions, while keeping the BSAMA concentration constant, the pore size of cryogel was found to be very different than hydrogels. Such an observation can be due to cryo-structuring (phase separation) via ice crystal (as a porogen) formation in the frozen phase, which can, in turn, facilitate the efficient cross-linking/polymerization in the unfrozen liquid microphase and lead to a gel with a large pore structure [44]. In contrast, the free radical polymerization of the randomly scattered monomer chains in a solution at room temperature seems to result in a gel with smaller pores with thick walls [50]. The mean pore size of hydrogels was smaller than that of cryogels; for example, 15% BSAMA hydrogels had a mean pore size of 25 µm whereases 15% BSAMA cryogels possessed an average pore size of 124 µm (Figure 3a,c). In general, the average pore size decreased with the increase in BSAMA concentration. In cryogelation, this result may be explained as increasing the polymer concentration can increase the cross-linking density, which may reduce the space available for the formation of ice crystals and consequently decrease the pore size. For hydrogels, an increase in polymer concentration can increase the crosslinking density and reduce main chain mobility, resulting in a significant reduction in pore size [51].

SEM, which may potentially compress and change the gel structure during the sample preparations, [52] may cause the variation in morphology. Therefore, to illustrate the porous character in a hydrated state, confocal microscopy was used via staining the gels with Rhodamine-B (Figure 3b). The transparent hydrogels exhibited uniform staining indicative of homogeneous gel matrix, allowing for high transmission of photosynthetically dynamic light wavelengths; their microporosity appeared not visible [47]. In contrast, the macroporous structure of BSAMA cryogels (Figure 3) was clearly visible. The opaque cryogels clearly demonstrated macroporosity resulting from phase separation [53]. The porous morphology and mean pore diameter of 10% BSAMA cryogels from CLSM results were close to those from SEM results. (Figure 3a,d).

The observed porous nature of the gels with interconnected pores in the range of 10 to 200 µm is significant because interconnected porous protein-based gels are highly preferred in biomaterial applications [54]. For instance, endothelial cells preferentially bind to scaffolds with smaller pores (≤80 µm), while fibroblasts bind favorably to bigger pores (>90 µm) [55]. Nutrients and oxygen can easily diffuse through such a scaffold with large pores whereas angiogenesis and cell migration are facilitated by the pore interconnectivity [56,57].

The swelling rate depends on pore distribution, size, interconnectivity, and orientation. A high equilibrium swelling rate for hydrogels and cryogels is essential when high absorption ability is compulsory for applications [58,59,60,61]. Consequently, the swelling behavior of the transparent and opaque BSAMA gels was tested (Figure 4a). It is evident that both of the gels have expanded to a maximum limit within two hours (Figure 4b–d). It is conceivable that the swelling rate depends on the preparation conditions and the concentration of the sample solution. The swelling rate of cryogels was higher than that of hydrogels, and the swelling rate was reduced by a high solution concentration. This result is consistent with the pore size distribution and pore size images by SEM (Figure 3). However, there is a phenomenon worth pondering. The swelling ratio of 10% (*w*/*v*) of BSAMA hydrogels is the largest among all samples (Figure 4b,e). It can be speculated that the low concentration and low crosslinking density of free radical polymerization at RT led to the formation of a hydrogel with a loose structure, hence the maximum swelling [62]. Moreover, the soft and structurally unstable nature of 10% BSAMA hydrogel can be witnessed in video S1. On the other hand, 10% BSAMA cryogels appeared structurally more stable than the counterpart 10% hydrogels because phase separation during cryogelation could cause BSAMA molecules to get close to one another, probably leading to improved crosslinking.

The mechanical strength of gels was investigated by the stress-strain curve of the samples gained through uniaxial compression testing (Figure 5). The cryogels could withstand greater deformation via compression as compared to hydrogels (Figure 5a). In contrast, hydrogels could not exhibit such mechanical properties. At the tested concentrations, hydrogels were more brittle than cryogels. As shown in Table 1, with the increase of concentration from 10% to 20%, the stress values at the breaking point demonstrating the mechanical strength of gels increased. Opaque gels illustrated a similar trend but with higher stress at break demonstrating high mechanical strength, as compared to hydrogels at the same polymer concentration. More importantly, cryogels demonstrated higher strain at break, which corresponds to the sponginess of the gels, compared to hydrogels under the same conditions: in addition, Young’s modulus was calculated by the slope below the strain value of 30% (Table 1). The modulus of hydrogels and cryogels depended on their concentration. The 15% hydrogels with a modulus of 38.95 ± 0.24 kPa were about 7.3 times higher than that of 10% (*w*/*v*) hydrogels (5.28 ± 0.10 kPa). The 15% cryogels with a modulus of 20.25 ± 0.001 kPa is 4.1 times higher than that of the 10% cryogel (4.92 ± 0.001 kPa). Hydrogels displayed a higher compressive modulus than cryogels at 10% and 15% concentrations. However, when the concentration reached 20%, the cryogels were stiffer than the hydrogels. In summary, the mechanical properties of the two gels are determined by their concentrations, and cryogels displayed better mechanical properties in terms of sponginess and mechanical strength (Figure 5b–d).

Tissue engineering is a technique for repairing, replacing, and regenerating damaged bone, cartilage, and skin tissues. Natural collagen and gelatin gels, for example, have been employed as materials for bone tissue engineering because of their unique biocompatibility and favorable physical features [63]. However, the required compressive strength and high fracture toughness of bone are mainly dependent on the nano-composite structure reinforced by hydroxyapatite crystals (stable at temperature up to 1200 °C) [64,65]. Bovine serum albumin decomposes in a temperature range of 320 to 575 °C [66]. To investigate the influence of methacrylation of BSA on thermal stability, TGA (Figure 6) was conducted by heating from 30 to 800 °C. Weight loss of BSA during TGA was consistent with previous studies. Figure 6 illustrates the thermal stability of BSA, BSAMA, BSAMA (15%)-based transparent hydrogel, and opaque cryogel. All samples presented equivalent initial decay between 30 to 150 °C with an ultimate thermal disintegration temperature of ~300 °C. More specifically, at the temperature range from 25 °C to 150 °C, the weight loss of the materials was around 8–10%, which corresponds to evaporation of some moisture. The materials began to degrade at about 220 °C, and the weight loss of BSA, BSAMA, transparent BSAMA gels and opaque BSAMA gels was 65%, 60%, 62%, and 60% respectively at 220–500 °C; methacryloylation of BSA has improved the thermal stability. Moreover, crosslinking by cryogelation makes BSAMA tougher and its thermal stability further improved [67,68]. This phenomenon is consistent with other BSAMA concentrations (10 and 20%) tested as illustrated in Appendix A.

It is necessary to investigate the biodegradability of biomaterials for their biomedical applications [33]. Proteinase K also known as protease K, has higher proteolytic action (Figure 7a) for natural proteins in physiological settings [69]. The degradation pattern of a material can be inclined through the material’s porosity. Degradation and permeabilization is directly influenced by the increase in porosity [70]. Other constraints like the homogeneity, size, and morphology of pores can also regulate the deterioration of porous constructs. [71] It was noted that the time needed for the degradation of hydrogels and cryogels was distinctly varied (Figure 7, Appendix A). 10% BSAMA hydrogels were completely decomposed within 2 h (Figure 7b), while 15% BSAMA hydrogels almost completely decomposed in 6 h (Figure 7c), and 20% BSAMA hydrogels lost about 70% at 6 h (Figure 7d). Whereas, 10% BSAMA cryogels maintained a weight of about 10% after 6 h (Figure 7b), and 15% and 20% BSAMA cryogels decreased 60% and 50% in weight after 6 h (Figure 7c and Figure 7d respectively). Interestingly, 15% and 20% opaque BSAMA cryogels had higher porosity than the counterpart hydrogels but showed a slower degradation rate. This further indicates that although the cryogels are more porous than hydrogels, the cryogels at sub-zero temperature freezing polymerization seem to be more tightly cross-linked than the RT polymerized hydrogels, hence leading to a slower degradation [62].

The capacity of cells to adhere, flourish, and differentiate is highly regulated via the porosity and distribution of the scaffold [72]. In order to determine the effect of hydrogels and cryogels on cell viability besides proliferation [73], 15% BSAMA was chosen for subsequent experiments. The skin of the elderly is easily injured and difficult to heal. This is probably due to the loss of fibroblasts in the upper skin. Fibroblasts can stimulate the growth of these cells and restore the elasticity of the skin. They can also stimulate the formation of hair follicles and reduce scars [74]. Based on these concepts, we used human skin fibroblasts (HSFs) to evaluate the cytocompatibility of hydrogel and cryogel scaffolds. HSFs were seeded and cultured on opaque and transparent gels over a period of 10 days (Figure 8a). As displayed in Figure 8b,c, all samples showed good cell compatibility. Cells attached to both hydrogels and cryogels and maintained viability throughout the culture period indicating that both transparent and opaque materials are suitable for cell culture [30]. Nevertheless, further research is necessary for a better explanation of these results by cell morphology, adherence, immunofluorescent staining, etc.

Previously, various platelet-rich plasma, hydroxyapatite sources, and antibacterial components have been incorporated into gels for the improvement of bone defects [75,76,77]. To illustrate such a potential application, an effective, feasible and tunable method of alternate soaking has been employed to mineralize the scaffolds [78]. As shown in Figure 9a, transparent and opaque gels in the course of mineralization time have become opaque demonstrating the mineral’s incessant growth. Later in the third cycle (day 3), both gels seemed entirely white, which demonstrates that the gel surface was almost covered with deposited minerals. SEM images of 15% BSAMA hydrogels (C0 and C5) and cryogels (C0 and C5) as shown in Figure 9b displayed that both hydrogels and cryogels demonstrated substantial deposition of mineral compared to pristine gels (C0).

As shown in Figure 9c,d, successful mineralization can be seen by the appearance of characteristic HA peaks (26°, and 32°) in the sequentially mineralized gels [79], whereas, pristine (C0) gels did not exhibit these peaks. Similarly, the presence of phosphate bands around 560 and 940 cm^−1^ further indicated the formation of hydroxyapatite in the gels (Figure 9e). The mineral content in the successively mineralized samples was quantified by TGA (Figure 9f). Because hydroxyapatite is thermally stable at temperatures up to 1200 °C, [64] the mineralized hydrogel samples were heated to 800 °C for complete removal with no hydroxyapatite loss. The percent remaining mass in hydrogel (C0) and cryogel (C0) was 28.681% and 28.816% around 580 °C. Pristine gels showed complete removal at 580 °C. Plateaus can be seen over a temperature range of 580–800 °C as an indication of no weight change for mineralized gels. Consequently, the percent remaining weight indicates the mineral content of the scaffolds at 800 °C. The mineral percentages for hydrogels were calculated as 33.381%, 53.382%, and 61.320%, for cycle 1,3, and 5 correspondingly, whereas, the mineral percentages of cryogel scaffolds were determined to be 36.818%, 56.648%, and 61.519%, for cycles 1,3, and 5, respectively. The results have shown a rise in mineral percentage with the increase in cycle number. In summary, both the hydrogels and cryogels are capable of forming HA-like minerals. We anticipate that HA (which is the major mineral constituent in natural bone [34]) mineralized gels can possibly be used for the preparation of biomimetic grafts aiming at bone tissue regeneration.

As seen in Figure 2, we propose the potential mechanism of the transparent and opaque gel formation. According to all the results obtained for transparent and opaque gels including specifically morphological structure (Figure 3), the smaller pores of hydrogels could be due to homogeneous crosslinking of generated radicals and solvated methacryloyl groups of BSAMA at RT [48]. RT gelation may cause the solvated monomer chains in the solution to be randomly crosslinked, resulting in the formation of a transparent gel matrix with small pores (Figure 3b) [53]. It can be speculated that this RT free radical polymerization of BSAMA can happen by chemical crosslinking as a result of the radical initiation (APS/TEMED as the initiator and catalyst), propagation of methacryloyl groups, and termination [80,81], while the solvation of BSAMA (mainly amide bonds) via hydrogen bonding with water may remain undisturbed during the radical polymerization. On the other hand, larger pores seen in cryogels could be due to cryogelation via phase separation by fast nucleation of ice crystals (as porogens) at a freezing temperature [82]. During the cryogelation, the ice crystal formation in the frozen phase and polymerization in the unfrozen monomer microphase can cause phase separation. As a result of phase separation and cryo-concentrated polymerization, the resulting cryogels can feature a large pore structure and opacity [44]. Therefore, it is proposed that different gelation mechanisms could elicit the transparency and opacity of the BSAMA gels (Figure 2).

## 3. Conclusions

We have successfully fabricated cost-effective transparent and opaque BSAMA gels with highly improved mechanical characteristics and controlled pore structures while not compromising the inherent protein structure. Compared with the corresponding BSAMA hydrogels prepared at room temperature, the BSAMA cryogels showed more enhanced physical and mechanical properties. BSAMA cryogels displayed more stable structural integrity and higher mechanical strength and deformability. A highly interconnected porous structure of opaque gels leads to an advanced fracture toughness [83]. It is presumed that different degrees of crosslinking may be achieved at different polymerization temperatures where different physical phenomena (solvation and phase separation) can take place and affect the properties of the final products. For instance, the cell proliferation rate of cryogels exceeded that of the corresponding hydrogels because cryogels provided more room for cell growth. Moreover, both the materials have shown good mineralizability and a different degree of biodegradability. To sum up, this study is mainly based on the facile development of BSAMA hydrogels and cryogels, which has strong innovation and scientific significance in terms of low cost, biocompatibility, biodegradability, improved mechanical properties, and macroporosity. This research will help to achieve the wider applications of the BSAMA-based gels in particularly in tissue repair.

## 4. Materials and Methods

### 4.1. Materials

Methacrylic anhydride (MAA, 94%), Bovine serum albumin (BSA), Deuterium oxide (D_2_O), Tris(hydroxymethyl)aminomethane, Proteinase K, Dulbecco’s phosphate-buffered saline (DPBS), and paraformaldehyde were purchased from Aladdin Chemistry Co., Ltd. (Shanghai, China). *N*, *N*, *N*′, *N*′-tetramethylethylenediamine (TEMED, ≥99.5%) and Ammonium persulfate (APS, >98%) have been bought by Shanghai yi en chemical technology Co., Ltd. (Shanghai, China). Human Skin Fibroblast (HSF) cells were purchased from Kunming Cell Bank, China. Trypsin-EDTA, Dulbecco’s modified Eagle’s medium (DMEM), fetal bovine serum (FBS), penicillin/streptomycin (P/S) were purchased from Gibco, Life Technologies, Beijing, China. Live/Dead Cell Viability/Cytotoxicity kit, and Rhodamine phalloidin (R415) were provided from Thermo Fisher Scientific Inc (Waltham, MA, USA). CCK8 kit was purchased by Dojindo Molecular Technologies (Fuzhou, China).

### 4.2. Methods

#### 4.2.1. Fabrication of BSAMA Biomaterials

Bovine Serum Albumin Methacryloyl (BSAMA) was synthesized following the earlier report [16]. DI water was used to prepare BSAMA with 10, 15, and 20% final *w*/*v* concentrations in the presence of 0.5% (*w*/*v*) ammonium persulfate (APS) and 0.25% (*w*/*v*) tetramethylethylenediamine (TEMED). Pre-cooled precursor solutions were transferred to yellow silicone rubber molds, and transferred to a freezer for 12 h at a pre-set sub-zero temperature (−80 °C) for the preparation of cryogels; they were also polymerized at room temperature (RT) to prepare hydrogels. After the reaction, the resulting frozen cryogels were brought to RT for thawing, and the final cryogels were obtained. Both hydrogels and cryogels were kept at 4 °C.

#### 4.2.2. Structural Characterization of BSA and BSAMA

Attenuated total reflection-Fourier transform infrared (ATR-FTIR) spectra of BSA, and BSAMA material, along with gels were carried out (4000–500 cm*^−^*^1^) by FTIR, TENSOR II, Germany. Following our earlier reports, [84,85] circular dichroism (CD) analysis was performed for BSA and BSAMA by Chirascan Plus (Applied Photophysics, Leatherhead, UK). The resulting spectra were smoothed and normalized and mean residue ellipticity (*MRE*; deg cm^2^ dmol*^−^*^1^) was calculated by the following equation:(1)$$MRE=\frac{\theta \;\times \;MRW}{l\;\times \;c}$$

*θ* = measured resulting ellipticity in mdeg,

*MRW* = mean residue weight,

*l* = path length in mm,

and *c* = protein concentration in mg mL*^−^*^1^.

MRE was used to calculate percent fractional helicity based on the *MRE* value at 208 nm (*θ*_208_) according to the literature report [86] and the following equation:
(2)$$Fractional\;helicity\;\%=\frac{\theta_{208}-\theta_{R}}{\theta_{\alpha }-\theta_{R}}\;\times \;100$$

*θ_R_* represents the *MRE* at 208 nm for protein with 100% random conformation and *θ_α_* shows the *MRE* at 208 nm for protein with 100% α-helix.

#### 4.2.3. Morphological Characterization

To analyze the morphology of gels in a dried condition, scanning electron microscopy (SEM, HITACHI, SU8010, Tokyo, Japan) was employed at 3 kV and 10 A. Hydrogel and cryogel samples with various concentrations were lyophilized and sliced in half from the center of the disc-shaped gels with 4.5 mm in height and 6.5 mm diameter. The gel slices were coated with a thin coating of platinum before the experiments. Each group was examined three times.

Furthermore, confocal microscopy was applied to visualize the porosity and internal structure of hydrated gels. The fully hydrated gels (three samples for each condition) were placed into a 24-well plate with 1 mg mL*^−^*^1^ Rhodamine B for confocal imaging. The Rhodamine B solution was incubated for 30 min in the dark, and the samples were rinsed with water until no pink color was visible. During the confocal laser scanning microscopy (CLSM, Nikon, A1), the samples were maintained hydrated. An excitation wavelength of 562 nm was used to create the images, and an emission wavelength of 573 nm was used to detect them.

For calculating pore sizes in the dry and wet state, three independent SEM and CLSM images of various hydrogels were employed to assess the mean pore diameter by ImageJ.

#### 4.2.4. Swelling Behavior

The prepared BSA-MA cryogels and hydrogels were soaked in DI water and incubated at 37 °C up to swelling equilibrium. First, the initial weight for each sample was recorded. Pre-weighed hydrogels and cryogels were immersed in deionized (DI) water at 37 °C and weighed at specified time points after gently removing surface water by Kimwipe (*m_i_*). After reaching the equilibrium swelling, the samples were weighed (*m_d_*)after freeze-drying. The following equation was used to measure the swelling rate:(3)$$Swelling\;rate\;\left(\%\right)=\frac{m_{i}-m_{d}}{m_{d}}\times 100$$

#### 4.2.5. Mechanical Properties

Compression tests at 90% strain were carried out to check the mechanical properties of the fully swollen cylindrical cryogels and hydrogels samples with a height of 4.5 ± 3 mm and a diameter of 6.5 ± 5 mm by mechanical testing machine (UTM2102; Shenzhen, China) at the speed of 10 mm/min. Three replicates were tested for each sample. Compressive modulus was obtained by using the slope of the stress-strain curve in the initial strain range of 20–30%.

#### 4.2.6. In vitro Enzymatic Degradation

In vitro enzymatic degradation of the as-prepared BSAMA cryogels and hydrogels was carried out by using a Proteinase K enzyme solution. Each sample was placed in PBS to reach equilibrium swelling, and after removing the moisture of each sample surface by Kim-wipe, its initial weight (*W_0_*) was measured. Then, 2 mL of a 0.1 mg mL*^−^*^1^. Proteinase K solution was added to each BSAMA cryogel and hydrogel and placed at 37 °C. At the designated time intervals (5, 10, 20, 30, 40, 50, 60, 120, 180, 240, and 300 min), the gel weight (*W_t_*) was sequentially measured. The residual rate (%) was calculated as follows:(4)$$Residual\;rate\;\left(\%\right)=\frac{W_{t}}{W_{0}}\;\times 100\;$$

#### 4.2.7. Cytocompatibility

Human Skin Fibroblast (HSF) cells were used as model cells to culture on BSAMA gels to evaluate cell compatibility. According to pre-experiments, 15% (w/v) BSAMA hydrogels and cryogels were selected to carry out the cell compatibility experiment. First, the cryogel and hydrogel samples were pretreated in a 48-well plate with a prepared medium (89% glucose (DMEM), 10% fetal bovine serum (FBS), and 1% antibiotic antifungal), and incubated at 37 °C and 5% carbon dioxide for 24 h. After that, 0.5 mL of HSF cell suspension (2 × 10^5^ mL^−1^) was inoculated onto each sample and cultured for 10 days, and the medium was changed every two days. Cells without gels in 48-well plates were used as a control group. Cell activity was determined using CCK-8 on 1, 4, 7, and 10 days. Cells with a CCK-8 solution (330 μL per sample; 1:10 CCK-8: medium) were incubated at 37 °C and 5% CO_2_ for 2 h. Absorbance was measured at 450 nm via the Infinity 200-Pro Microplate Reader. The relative vitality was calculated as follows:(5)$$Cell\;viability\;\left(\%\right)=\frac{{\left[A\right]}_{test}}{{\left[A\right]}_{control}}\;\times 100\;$$

[*A*]*_test_* is the absorbance of the test group; [*A*]*_control_* the absorbance of the control group.

HSF cells proliferation on hydrogels and cryogels over a period of 1, 4, 7, and 10 days was examined by Live/Dead Cell Viability, Cytotoxicity kit according to the manufacturer’s instructions. Cells were grown on 3D transparent hydrogels and opaque cryogels via the same process as mentioned above, and culture media were refreshed every two days. Briefly, green-colored live cells were stained with calcein-acetomethoxy (calcein-AM~4 μM), while ethidium homodimer-1 (EthD-1) at a concentration of 8 μM was used to stain dead cells with red color. Washing with PBS was done after 30-min, and confocal microscopy was used for imaging.

#### 4.2.8. Mineralization of Transparent and Opaque Gels

Tris-buffer was used for the dissolution of CaCl_2_ and Na_2_HPO_4_ to prepare 0.4 M calcium chloride (CaCl_2_) dihydrate and 0.24 M Dibasic Sodium Phosphate (Na_2_HPO_4_) solutions (pH was maintained at 7.5 by using HCl), respectively. Each of the 15% transparent and opaque BSAMA gel disks was incubated in a 1 mL CaCl_2_ for 12 h and washed with DI water. Later, discs were incubated in 1 mL Na_2_HPO_4_ for 12h with subsequent rinsing in DI. This incubation period of 24 h was termed one cycle. Subsequently, D1, D3 and D5 mineralized scaffolds were obtained. An autoclave was used to sterilize the solutions of BSAMA, DI water, CaCl_2_, and Na_2_HPO_4_.

#### 4.2.9. X-ray Diffraction (XRD) Investigation:

Mineralized hydrogels were tested for XRD analysis using a Bruker D8 ADVANCE (Germany). XRD data of finely powdered lyophilized gels were obtained for 2*θ* = 50°.

#### 4.2.10. Thermogravimetric Analysis (TGA)

Thermal stability for BSAMA-based transparent hydrogels and opaque cryogels was measured by Perkin Elmer Pyris Dimond model TGA 4000. 10 °C min*^−^*^1^ heating rate was used to heat 10 mg of each sample over a range of temperature from 30 to 800 °C under nitrogen atmosphere (N_2_ flow: 50 cm^3^ min*^−^*^1^).

#### 4.2.11. Statistical Analysis

OriginPro 9.800200 and GraphPad Prism 6.01.298 was used for statistical analysis. Triplicate specimens of the individual case were utilized, and results were shown as mean ± standard deviation (SD). Independent *t*-tests were used for the statistical significance of the results, and the *p*-values < 0.05 were considered statistically significant.

## Data Availability

The data generated from the study is clearly presented and discussed in the manuscript.

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
