# Peer review of "Facile Fabrication of Transparent and Opaque Albumin Methacryloyl Gels with Highly Improved Mechanical Properties and Controlled Pore Structures"

_gels, 2022, doi:10.3390/gels8060367_

Round 1

Reviewer 1 Report

Thank you for the good job. Please, cite the article below, as well.

https://www.nature.com/articles/srep19370

Author Response

Reviewer 1

Comments and Suggestions for Authors

Thank you for the good job. Please, cite the article below, as well.

https://www.nature.com/articles/srep19370

 Response: We are highly thankful to the reviewer for kind remarks. Furthermore, we have cited the relevant research article (ref # 5 in the revised manuscript) as advised.

Reviewer 2 Report

The manuscript “Facile fabrication of transparent and opaque albumin methacryloyl gels with highly improved mechanical properties and controlled pore structures” develops a Bovine serum albumin (BSA) based gel system for tissue engineering application. The authors demonstrated meticulous steps of optimization and characterizations of such BSA-based gel system, with two polymerization temperature (room temperature and -80 °C). They also demonstrate a practical utility of the system, as bone tissue engineering scaffolds with successful mineralization. The work is interesting and precedent, although it is preliminary.

I think it is acceptable after some revision, taking into account the following points.

Major points:

  1. The system presented is novel, towards a tissue engineering application. Figure 6 demonstrated thermal stability of such system, with temperature up to 800 °C. Throughout the paper, the authors emphasis is on eventual tissue engineering application, which typically does not reach such high temperature. Please provide additional context to justify including such thermal stability characterization in this paper.
  2. Figure 7 demonstrated the system’s biodegradability, with experimental details in Section 4.2.6 (Line 429). What is the starting weight of the gel in such experiment? With such information, it will help readers figure out the weight ratio between Protease K and gel.

Minor points:

  1. Please check the sentence in Line 328 “In summary, together hydrogels too cryogels are capable of forming HA-like minerals.” The wording “together” “too” made it hard to understand this sentence.
  2. Line 369 “super porosity”. This is not a well-defined term. Please provide additional explanation/ definition, or clarify what the authors want to convey.

Author Response

Thank you much for your comments and suggestions.

Reviewer 3 Report

see attachment

Author Response

Thank you so much for your insightful and critical comments and suggestions. We have provided a point-by-by response to each comment. Please see the attachment.

Round 2

Reviewer 3 Report

my concerns were adequately addressed, paper is ready for publication